# Multicentre Surveillance of *Candida* Species from Blood Cultures during the SARS-CoV-2 Pandemic in Southern Europe (CANCoVEU Project)

**DOI:** 10.3390/microorganisms11030560

**Published:** 2023-02-23

**Authors:** Matteo Boattini, Margarida Feijó Pinto, Eirini Christaki, Teresa Fasciana, Iker Falces-Romero, Andreas Tofarides, Gabriele Bianco, Emilio Cendejas-Bueno, Maria Rita Tricoli, Giorgos Tsiolakkis, Julio García-Rodríguez, Rafail Matzaras, Sara Comini, Anna Giammanco, Diamanto Kasapi, André Almeida, Konstantina Gartzonika, Rossana Cavallo, Cristina Costa

**Affiliations:** 1Microbiology and Virology Unit, University Hospital Città della Salute e della Scienza di Torino, 10126 Turin, Italy; 2Department of Public Health and Paediatrics, University of Torino, 10124 Turin, Italy; 3Serviço de Patologia Clínica, Laboratório de Microbiologia, Centro Hospitalar Universitário de Lisboa Central, 1169-45 Lisbon, Portugal; 4Department of Internal Medicine and Infectious Diseases Unit, University Hospital of Ioannina, 455 00 Ioannina, Greece; 5Department of Health Promotion, Mother and Child Care, Internal Medicine and Medical Specialties, University of Palermo, 90127 Palermo, Italy; 6Clinical Microbiology and Parasitology Department, Hospital Universitario La Paz, Paseo de la Castellana, 261, 28046 Madrid, Spain; 7CIBERINFECT, Instituto de Salud Carlos III, 28046 Madrid, Spain; 8Department of Internal Medicine, Nicosia General Hospital, Nicosia 2029, Cyprus; 9Department of Internal Medicine 4, Hospital de Santa Marta, Central Lisbon Hospital Centre, 1169-050 Lisbon, Portugal; 10NOVA Medical School, Universidade Nova de Lisboa, Campo dos Mártires da Pátria 130, 1169-056 Lisbon, Portugal; 11Department of Microbiology, Faculty of Medicine, University of Ioannina, 451 10 Ioannina, Greece

**Keywords:** *Candida* spp., candidaemia, blood culture, COVID-19, bloodstream infection, fungaemia, SARS-CoV-2

## Abstract

**Introduction**: Surveillance of *Candida* species isolates from blood cultures (BCs) in Europe is considered fragmented, unable to allow the definition of targets of antifungal stewardship recommendations especially during the SARS-CoV-2 pandemic. **Methods**: We performed a multicentric retrospective study including all consecutive BC *Candida* isolates from six Southern European tertiary hospitals (1st January 2020 to 31st December 2021). Etiology, antifungal susceptibility patterns, and clinical setting were analyzed and compared. **Results**: *C. albicans* was the dominant species (45.1%), while *C. auris* was undetected. *Candida* species positive BC events increased significantly in COVID-19 ICUs in 2021 but decreased in other ICUs. Resistance to azole increased significantly and remained very high in *C. albicans* (fluconazole from 0.7% to 4.5%, *p* = 0.03) and *C. parapsilosis* complex (fluconazole up to 24.5% and voriconazole up to 8.9%), respectively. Resistance to caspofungin was remarkable in *C. tropicalis* (10%) and *C. krusei* (20%), while resistance to at least one echinocandin increased in 2021, especially in *C. parapsilosis* complex (from 0.8% to 5.1%, *p* = 0.05). Although no significant differences were observed over the study period, fluconazole and echinocandin resistance increased in COVID-19 ICUs by up to 14% and 5.8%, respectively, but remained undetected in non-intensive COVID-19 wards. **Conclusions**: Antifungal stewardship activities aimed at monitoring resistance to echinocandin in *C. tropicalis* and *C. krusei*, and against the spread of fluconazole resistant *C. parapsilosis* complex isolates are highly desirable. In COVID-19 patients, antifungal resistance was mostly present when the illness had a critical course.

## 1. Introduction

Candidemia is one of the most frequent health care-associated bloodstream infections (BSIs) and represents a global clinical challenge, especially given the burden of associated morbidity and mortality [1,2]. Several authors have reported an increase in the incidence of *Candida* species BSIs during the SARS-CoV-2 pandemic, highlighting the need for active surveillance especially in patients with severe COVID-19 [3,4,5,6,7]. Antibiotic therapy, corticosteroids, immunosuppressive therapy, intravascular devices, long hospital stays, and direct disruption of the intestinal barrier caused by SARS-CoV-2 have been deemed to pave the way to *Candida* species BSIs [4,5]. Despite the implementation of infection control measures during the pandemic, several authors reported cases of infections due to the emerging multidrug-resistant *C. auris* [8,9,10], highlighting flaws in antimicrobial stewardship programs probably due to hospital reorganization, use of broad-spectrum antimicrobials and horizontal spread of resistant strains [11]. 

Despite their suboptimal sensitivity, blood cultures (BCs) remain the diagnostic reference standard for *Candida* species BSIs, as they allow for both the identification of *Candida* at the species level and susceptibility testing. *Candida* species identification is of paramount importance to promptly optimize antifungal therapy, safeguard the use of more expensive antifungals, de-escalate treatment whenever possible and steer antifungal stewardship efforts. In fact, in addition to *C. krusei* being intrinsically resistant, resistance to fluconazole was also reported to be remarkable only for *C. glabrata* [12], *C. guilliermondii* [13], *C. auris* [14], and, more recently, in *C. parapsilosis* complex [6,7,15]. 

Data on the impact of SARS-CoV-2 pandemic on the epidemiology of *Candida* species BSIs and resistance to antifungal agents in multicenter surveillance studies are limited. Likewise, surveillance of antifungal resistance in *Candida* species BC isolates in Europe is considered fragmented, unable to define the burden and proper targets of antifungal stewardship recommendations [16]. This study was aimed at monitoring and comparing the epidemiology and antifungal susceptibility of *Candida* species isolated from BCs collected from patients admitted in six Southern European tertiary hospitals during the first two years of the COVID-19 pandemic.

## 2. Materials and Methods

### 2.1. Study Design

We performed a multicentric retrospective observational study including all consecutive BC *Candida* isolates from six European tertiary hospitals located in five countries (Northern Italy, Southern Italy, Portugal, Spain, Greece, and Cyprus; 6225 hospital beds overall) collected from 1st January 2020 to 31st December 2021. BCs yielding the same *Candida* species and subsequent samples obtained within 20 days of each other were regarded as representing a continuing positive BC event and were therefore excluded from the analysis.

### 2.2. Aims of the Study

The primary aim was to depict the epidemiology of the *Candida* species isolated from BCs of a European cohort of patients hospitalized during the first two years of the SARS-CoV-2 pandemic. The secondary aim was to evaluate the antifungal susceptibility patterns of the *Candida* species included.

### 2.3. Candida Species Identification and Susceptibility

For each *Candida* species positive BC event, the following data were recorded: clinical setting in which the pathogen was isolated (emergency, medical, surgical, or COVID-19 wards, ICU, COVID-19 ICU) and susceptibility testing results. Species identification method, antifungal susceptibility testing, and clinical breakpoints used by the respective institution during the study period were considered (Appendix A).

Isolation of the *Candida* species was mostly performed using RPMI 1640 and/or BBLTM CHROMagarTM Candida medium (Becton Dickinson GmbH, Heildelberg, Germany).

*Candida* species identification was performed using Vitek 2 (Biomérieux, Mercy l’Ètoile, France), for biochemical identification, Vitek MS (Biomérieux, Mercy l’Ètoile, France) or Bruker Biotyper (Bruker DALTONIK GmbH, Bremen, Germany), for MALDI-TOF mass spectrometry-based identification. 

Susceptibility testing results were obtained using broth microdilution commercial systems (SensititreTM YeastOneTM, TREK Diagnostic Systems, Cleveland OH; MICRONAUT-AM Antifungal Agents MIC, MERLIN Diagnostika GmbH Systems, Bornheim, Germany; Vitek 2, Biomériéux, Mercy l’Ètoile, France), gradient test (Etest, Biomérieux, Mercy l’Ètoile, France), and disc diffusion (Liofilchem, Roseto degli Abbruzzi, Italy), according to the recommendations of the respective manufacturers.

Results of the antimicrobial susceptibility testing were interpreted according to the European Committee on antimicrobial susceptibility testing (EUCAST. v. 10.0; www.eucast.org) or Clinical & Laboratory Standards Institute (CLSI. *Performance Standards for Antifungal Susceptibility Testing of Yeasts.* 3rd ed. CLSI supplement M27M44S (Suppl.4). Clinical and Laboratory Standards Institute (CLSI); www.clsi.org) guidelines.

Resistance to azole was defined as resistance to at least one antifungal agent among fluconazole and voriconazole. Resistance to echinocandin was defined as resistance to at least one antifungal agent among anidulafungin, micafungin, and caspofungin.

### 2.4. Statistics

Descriptive data were shown as absolute (*n*) and relative (%) frequencies for categorical data. The chi-square test or Fisher exact test were used to compare the distribution of the categorical variables. Summary statistics used for MIC values included the MIC50 and MIC90. For all tests, a *p* value ≤ 0.05 was considered significant. All analyses will be performed with SPSS v. 25.0 (IBM Corp., Armonk, NY, USA).

## 3. Results

In the two-year period, 250,591 BCs were processed. Among these, 1451 were positive for *Candida* species and 745 deemed to be *Candida* species positive BC events and suitable to be analyzed for the aims of the study (Table 1). The most frequently identified *Candida* species were *C. albicans* (45.1%), *C. parapsilosis* complex (31.8%), *C. glabrata* (14.1%), and *C. tropicalis* (5.4%). The comparison of the *Candida* species distributions over the study period showed a statistically significant increase in *C. albicans* (*p* = 0.02) and *C. krusei* (*p* = 0.03) and a decrease in *C. parapsilosis* complex (*p* < 0.01) in 2021. 

The analysis of the yearly distribution of *Candida* species according to hospital ward (Table 2) showed that *Candida* species BC events occurred more frequently in medical wards (34.9%), followed by ICUs (23.6%), and surgical wards (21.6%). Moreover, *Candida* species BC events decreased in non-COVID-19 ICUs (*p* = 0.02) and increased significantly in COVID-19 ICUs (*p* = 0.02) in 2021. The comparison of the distribution of *Candida* species over the two years according to hospital ward (Appendix A) showed that in 2021 (1) *C. albicans* increased in both surgical wards (*p* = 0.03) and ICUs (*p* = 0.05), while *C. parapsilosis* complex decreased in the same departments (*p* = 0.02 and *p* < 0.01, respectively); (2) *C. glabrata* increased in medical wards.

The results of the antifungal susceptibility testing (Table 3) showed (1) resistance to azole and echinocandin in *C. albicans* were <5% and <0.7%, respectively; (2) resistance to fluconazole, voriconazole, and echinocandin in *C. parapsilosis* complex were EUCAST 31.2% vs. CLSI 11.7%, EUCAST 11% vs. CLSI 0%, and <5%, respectively; (3) resistance to fluconazole, micafungin, and anidulafungin in *C. glabrata* were EUCAST 10.4% vs. CLSI 13.5%, CLSI 5.7%, and CLSI 6.3%, respectively; (4) resistance to fluconazole, voriconazole, caspofungin, and anidulafungin in *C. tropicalis* were EUCAST 0% vs. CLSI 6.7%, EUCAST 0% vs. CLSI 6.7%, EUCAST 10% vs. CLSI 0%, and EUCAST 10% vs. CLSI 12.5%, respectively; (5) resistance to caspofungin and anidulafungin in *C. krusei* were EUCAST 20% vs. CLSI 0% and EUCAST 25% vs. CLSI 0%, respectively; (6) resistance to fluconazole, caspofungin, and micafungin in *C. guilliermondii* were CLSI 33.3% for each antifungal.

The comparison of azole and echinocandin resistance rates between 2020 and 2021 (Table 4) showed that there were (1) no statistically significant increases in resistance to azole and echinocandin, even excluding *C. krusei*; (2) a statistically significant increase in resistance to fluconazole by up to 4.5% in *C. albicans* (*p* = 0.03); (3) an increase in resistance to azole (fluconazole up to 24%; voriconazole up to 5.3%) and echinocandin (up to 4.9%) in non-*albicans Candida*; (4) a high and persistent resistance rate to azole, especially fluconazole (range 22–24.5%), in *C. parapsilosis* complex; (5) a statistically significant increase in resistance to echinocandin (up to 5.1%; *p* = 0.05) in *C. parapsilosis* complex and a reduction in both azole and echinocandin resistance rates in *C. glabrata*. The comparison of azole and echinocandin resistance rates according to hospital ward (Appendix A) showed no statistically significant difference over the study period. However, an increase in azole and echinocandin resistance occurred in COVID-19 ICUs, but not in non-intensive COVID-19 wards.

## 4. Discussion

This study investigated the epidemiology of *Candida* species isolated from BCs in a European cohort of patients hospitalized during the first two years of the SARS-CoV-2 pandemic, a period of incessant hospitals reorganization, high patient care load and huge antimicrobial prescription. Our results showed that *Candida* species BC events most frequently occurred in medical wards, *C. albicans* was the dominant species, *C. auris* was undetected, and while *C. parapsilosis* complex decreased in 2021, *C. krusei* increased significantly. Overall, *Candida* species positive BC events increased significantly in COVID-19 ICUs and decreased in the other ICUs in 2021. Overall antifungal resistance was low among the *Candida* species isolates tested, except for *C. parapsilosis* complex (fluconazole and EUCAST voriconazole), *C. glabrata* (fluconazole), *C. tropicalis* (anidulafungin and EUCAST caspofungin), *C. krusei* (EUCAST caspofungin and EUCAST anidulafungin), and *C. guilliermondii* (CLSI fluconazole, CLSI caspofungin, and CLSI micafungin) where resistance ≥10% (range 10–33.3%) to one or more antifungal agents was observed. Resistance rate to azole over the study period increased significantly in *C. albicans* (fluconazole) and remained very high in *C. parapsilosis* complex. Resistance to echinocandin increased in 2021, especially in *C. parapsilosis* complex, while it was undetected in both *C. albicans* and *C. glabrata* during the same period. Although no significant differences were shown over the study period, azole and echinocandin resistance increased in COVID-19 ICUs while these remained undetected in non-intensive COVID-19 wards.

Although the distribution of *Candida* species varies according to geographic areas and may be related to age, level of care intensity, and prior use of antifungals, *C. albicans* is the most frequently isolated from the BCs and, together with *C. parapsilosis* complex and *C. glabrata*, have been reported as the etiological agents of more than 90% of *Candida* species BSIs [16,17,18]. Our findings are consistent with this evidence and, as well as highlight the burden of *Candida* species positive BC events in medical wards [19], confirm that *C. auris* is currently identified from BCs only sporadically or during nosocomial outbreaks [8,9,10,11]. However, it must be emphasized that for ecological reasons related to the colonization of patients and hospital environments, a longer study time may be necessary to document possible future positive BC events caused by *C. auris*. Therefore, our data suggest that, at present, the SARS-CoV-2 pandemic does not appear to have resulted in dramatic epidemiological changes in *Candida* species, although a marginal increase in *C. krusei* is worth monitoring. Of note, the 2021 increase in *Candida* species positive BC events in COVID-19 ICUs, with a matching decrease in other ICUs. This finding could be linked, in addition to the increased incidence of *Candida* species BSI in patients with severe COVID-19, to hospital reorganization, which was undertaken to increase availability of COVID-19 ICU beds, frequently by converting pre-existing ICU beds or using other non-state-of-the-art-ICU facilities.

Global 2020 data on CLSI susceptibility testing results of clinical *Candida* species isolates presented by the ARIA (Analysis of Resistance In Antifungals) surveillance initiative [20] recently showed reduced susceptibility to both fluconazole and voriconazole in *C. auris*, remarkable resistance to caspofungin in *C. glabrata* (99.2%), *C. krusei* (87.3%), and *C. tropicalis* (19.8%), and non-negligible resistance to azole in *C. parapsilosis* complex (fluconazole 9.5% and voriconazole 7.8%). Similarly, European 2019 data elaborated from 252 *Candida* species isolates from the Czech Republic, Germany, Italy, and Turkey [21] showed relevant resistance to azole in *C. parapsilosis* complex (fluconazole 33.3% and voriconazole 20%), remarkable resistance to voriconazole (14.7%) and echinocandins (anidulafungin 57.4% and caspofungin 10.3%) in *C. glabrata*, remarkable resistance to echinocandins (caspofungin 19% and micafungin 14.3%) in *C. krusei*, and remarkable resistance to fluconazole (12.5%) and anidulafungin (12.5%) in *C. guilliermondii*. 

Although the agreement between the commercial and reference methods for *Candida* species susceptibility has been described as variable, since it may depend on the antifungals, the species, and the incubation time [22,23,24,25], our results provided relevant indications. In fact, the remarkable resistance rates to: 1) fluconazole in *C. parapsilosis* complex (up to 31.2%), *C. glabrata* (up to 13.5%), and *C. guilliermondii* (up to 33.3%); 2) EUCAST voriconazole in *C. parapsilosis* complex (up to 11%); 3) EUCAST and/or CLSI echinocandin in *C. tropicalis* (up to 12.5%), *C. krusei* (up to 25%), and *C. guilliermondii* (up to 33.3%) were consistent with those of both the ARIA project and recent reports analyzing clinical samples from Greece and the Madrid region that highlighted the emergence of fluconazole resistance in *C. parapsilosis* complex (up to 23%) and a low rate of resistance to echinocandins (up to 1–3%) [7,15]. Our results went into further detail by also highlighting an increase in resistance to fluconazole in *C. albicans* (up to 4.5%), and fluconazole (up to 24.5%) and echinocandin (up to 5.1%) in *C. parapsilosis* complex, respectively, despite the reduced frequency of *C. parapsilosis* complex in 2021. It is also important to point out that a difference in resistance rates to certain antifungals has been found in some *Candida* species depending on the type of breakpoints used. In fact, in our study, this was more evident especially for fluconazole (EUCAST 31.2% vs. CLSI 11.7%) and voriconazole (EUCAST 11% vs. CLSI 0%) resistance in *C. parapsilosis* complex, and caspofungin (EUCAST 20% vs. CLSI 0%) and anidulafungin (EUCAST 25% vs. CLSI 0%) resistance in *C. krusei*. Even though in need of harmonization, EUCAST and CLSI methods are described to produce comparable results for testing these agents against the five most common species of *Candida* [26,27]. One possible explanation for this discrepant data could be that the dissemination of fluconazole- and/or voriconazole-resistant *C. parapsilosis* complex is still restricted to certain geographical areas and occurred predominantly and completely by chance in areas belonging to centers using the EUCAST method. The same speculation could be made for *C. krusei*, but the limited number of isolates tested prevents this. Of note, the fact that azole and echinocandin resistance increased in COVID-19 ICUs and remained undetected in non-intensive COVID-19 wards might emphasize that while *Candida* species BSIs may have many predisposing factors, antifungal resistance is mostly present when the illness has a critical course [28].

The main strength of this study is its large number of BC isolates across different countries, which allowed better assessment of the epidemiology of *Candida* species and antifungal resistance in Southern Europe.

The limited number of isolates limits the generalizability of results regarding *C. tropicalis*, *C. krusei*, and *C. guilliermondii* and should be considered a study limitation. In addition, the heterogeneity of the SARS-CoV-2 pandemic and the relative diagnostic capacities in the different centers even belonging to the same countries in the early 2020s and the continuous hospital reorganizations may have had a small influence on some of the results presented regarding the 2020 vs. 2021 epidemiological comparisons.

## 5. Conclusions

Our investigation showed an epidemiological picture in which *C. albicans* remained the main *Candida* species in positive BC events during the SARS-CoV-2 pandemic. No *C. auris* positive BCs were detected. Most *Candida* species BC events occurred in medical wards. The most remarkable finding was the resistance rate to fluconazole that increased significantly by up to 4.5% and remained >22% in *C. albicans* and *C. parapsilosis* complex, respectively. Resistance to echinocandin, which was generally low, increased in 2021, especially in *C. parapsilosis* complex. Hospital reorganization and conversion of ICUs into COVID-19 ICUs might have contributed to the numerical shift in *Candida* species positive BC events in these units. However, the increase of these events in COVID-19 ICUs was accompanied by an increase in resistance to azole and echinocandin, which was not the case in non-intensive COVID-19 wards. Further surveillance studies are warranted to confirm these findings and design antifungal stewardship activities, especially against the spread of fluconazole resistant *C. parapsilosis* complex isolates.

## Figures and Tables

**Table 1 microorganisms-11-00560-t001:** Distribution per year of *Candida* species positive blood culture events.

Year	Quarter	*Candida* Species Positive BC Event, *n*	*C. albicans* % (*n*)	*C. parapsilosis* Complex % (*n*)	*C. glabrata* % (*n*)	*C. tropicalis* % (*n*)	*C. krusei* % (*n*)	*C. guilliermondii* % (*n*)	*C. lusitaniae* % (*n*)	*C. dubliniensis* % (*n*)
2020	1°	90	40 (36)	28.9 (26)	16.7 (15)	10 (9)	2.2 (2)	1.1 (1)	1.1 (1)	-
	2°	97	37.1 (36)	49.4 (48)	9.3 (9)	3.1 (3)	-	-	1.1 (1)	-
	3°	107	43 (46)	36.5 (39)	13.1 (14)	5.6 (6)	0.9 (1)	-	-	0.9 (1)
	4°	86	44.2 (38)	34.8 (30)	12.8 (11)	3.5 (3)	-	2.3 (2)	1.2 (1)	1.2 (1)
Subtotal		380	41.1 (156)	37.6 (143)	12.9 (49)	5.5 (21)	0.8 (3)	0.8 (3)	0.8 (3)	0.5 (2)
2021	1°	73	56.2 (41)	20.5 (15)	19.2 (14)	2.7 (2)	-	1.4 (1)	-	-
	2°	101	46.5 (47)	20.8 (21)	19.8 (20)	5.9 (6)	5 (5)	1 (1)	1 (1)	-
	3°	99	44.4 (44)	32.3 (32)	10.1 (10)	8.1 (8)	3 (3)	2.1 (2)	-	-
	4°	92	52.3 (48)	28.3 (26)	13 (12)	3.2 (3)	3.2 (3)	-	-	-
Subtotal		365	**49.3** (180)	**25.8** (94)	15.3 (56)	5.2 (19)	**3** (11)	1.1 (4)	0.3 (1)	-
Total		745	45.1 (336)	31.8 (237)	14.1 (105)	5.4 (40)	1.9 (14)	0.9 (7)	0.5 (4)	0.3 (2)

Abbreviations: BC: blood culture. Numbers in bold indicate statistically significant difference (*p* ≤ 0.05).

**Table 2 microorganisms-11-00560-t002:** Distribution per year of *Candida* species according to hospital ward.

*Candida* Species	Emergency % (*n*)	Medical Ward % (*n*)	Surgical Ward % (*n*)	ICU % (*n*)	COVID-19 ICU % (*n*)	COVID-19 Ward % (*n*)
*C. albicans n* = 336	4.5 (15)	33.3 (112)	23.2 (78)	22 (74)	13.7 (46)	3.3 (11)
*C. parapsilosis* complex *n* = 237	1.7 (4)	37.9 (90)	19 (45)	27 (64)	12.7 (30)	1.7 (4)
*C. glabrata n* = 105	7.6 (8)	37.1 (39)	21.9 (23)	22.9 (24)	6.7 (7)	3.8 (4)
*C. tropicalis n* = 40	10 (4)	35 (14)	27.5 (11)	7.5 (3)	17.5 (7)	2.5 (1)
*C. krusei n* = 14	7.1 (1)	14.3 (2)	-	57.2 (8)	21.4 (3)	-
*C. guilliermondii n* = 7	-	-	42.8 (3)	28.6 (2)	28.6 (2)	-
*C. lusitaniae n* = 4	-	25 (1)	25 (1)	25 (1)	25 (1)	-
*C. dubliniensis n* = 2	-	100 (2)	-	-	-	-
Subtotal 2020 *n* = 380	3.4 (13)	37.4 (142)	20 (76)	27.1 (103)	10 (38)	2.1 (8)
Subtotal 2021 *n* = 365	5.2 (19)	32.3 (118)	23.3 (85)	**20** (73)	**15.9** (58)	3.3 (12)
Total	4.3% (32)	34.9% (260)	21.6% (161)	23.6% (176)	12.9% (96)	2.7% (20)

Abbreviations: ICU: intensive care unit. Numbers in bold indicate statistically significant difference (*p* ≤ 0.05).

**Table 3 microorganisms-11-00560-t003:** Susceptibility testing results of *Candida* species isolates tested in the study.

		*Candida albicans* *n* = 336			*Candida parapsilosis* Complex *n* = 237			*Candida glabrata* *n* = 105			*Candida tropicalis* *n* = 40		
Antifungal Agent		MIC_50_ (mg/L)	MIC_90_ (mg/L)	% Resistance (*n*)	MIC_50_ (mg/L)	MIC_90_ (mg/L)	% Resistance (*n*)	MIC_50_ (mg/L)	MIC_90_ (mg/L)	% Resistance (*n*)	MIC_50_ (mg/L)	MIC_90_ (mg/L)	% Resistance (*n*)
Fluconazole	EUCAST	0.5	1	2.4 (5/205)	4	12	31.2 (44/141)	4	16	10.4 (6/58)	1	2	(0/24)
	CLSI	0.25	1	3.3 (4/121)	0.5	8	11.7 (10/94)	4	16	13.5 (5/37)	1	1	6.7 (1/15)
Voriconazole	EUCAST	0.078	0.12	4.9 (9/185)	0.12	0.5	11 (15/137)	0.06	0.25	-	0.125	0.125	(0/24)
	CLSI	0.015	0.12	1.7 (2/120)	0.016	0.25	(0/94)	0.12	0.25	5.9 (2/34)	0.12	0.12	6.7 (1/15)
Posaconazole	EUCAST	0.078	0.015	(0/90)	0.015	0.06	4 (2/50)	0.5	1	-	0.015	0.06	(0/10)
	CLSI	0.03	0.06	-	0.03	0.12	-	0.25	1	-	0.06	0.25	-
Itraconazole	EUCAST	0.031	0.047	2.6 (4/153)	0.03	0.125	4.7 (3/64)	0.5	4	-	<0.007	<0.007	(0/10)
	CLSI	0.03	0.12	(0/68)	0.03	0.12	(0/54)	0.25	0.5	2.9 (1/34)	0.12	0.25	12.5 (1/8)
Isavuconazole	EUCAST	-	-	-	-	-	-	-	-	-	-	-	-
	CLSI	≤0.008	0.015	(0/44)	≤0.008	≤0.008	(0/23)	0.03	0.12	(0/17)	≤0.008	0.25	16.7 (1/6)
Caspofungin	EUCAST	0.06	0.125	0.7 (1/140)	0.5	1	(0/101)	0.015	0.015	2 (1/51)	0.125	0.125	10 (1/10)
	CLSI	0.06	0.25	(0/119)	0.5	1	4.3 (4/94)	0.03	0.25	(0/37)	0.12	0.25	(0/15)
Micafungin	EUCAST	0.015	0.015	0.7 (1/136)	0.25	0.5	(0/117)	0.015	0.015	(0/51)	0.016	0.12	-
	CLSI	0.008	0.016	(0/95)	0.5	1	2.5 (2/80)	0.016	0.016	5.7 (2/35)	0.06	0.12	8.3 (1/12)
Anidulafungin	EUCAST	0.015	0.015	0.6 (1/160)	0.25	3	1.6 (1/63)	0.015	0.031	1.9 (1/53)	0.031	0.125	10 (1/10)
	CLSI	0.05	0.12	(0/68)	1	2	(0/53)	0.015	0.03	6.3 (2/31)	0.12	0.25	12.5 (1/8)
Amphotericin B	EUCAST	0.25	1	(0/209)	0.5	1	5 (7/141)	0.5	1	(0/67)	0.25	0.5	(0/24)
	CLSI	0.5	1	-	0.5	1	-	0.5	1	-	0.5	1	-
		** *Candida krusei* ** ***n* = 14**			** *Candida guilliermondii* ** ***n* = 7**			** *Candida lusitaniae* ** ***n* = 4**			** *Candida dubliniensis* ** ***n* = 2**		
		**MIC_50_ (mg/L)**	**MIC_90_ (mg/L)**	**% resistance (*n*)**	**MIC_50_ (mg/L)**	**MIC_90_ (mg/L)**	**% Resistance (*n*)**	**MIC_50_ (mg/L)**	**MIC_90_ (mg/L)**	**% Resistance (*n*)**	**MIC_50_ (mg/L)**	**MIC_90_ (mg/L)**	**% Resistance (*n*)**
Fluconazole	EUCAST	-	-	-	-	-	-	1	1	(0/1)	0.25	0.25	(0/1)
	CLSI	32	32	-	2	4	33.3 (1/3)	1	1	(0/2)	0.25	0.25	(0/1)
Voriconazole	EUCAST	≤0.12	≤0.12	-	-	-	-	-	-	-	0.016	0.016	(0/1)
	CLSI	0.12	0.12	(0/5)	<0.12	<0.12	(0/6)	0.015	0.06	(0/2)	0.008	0.008	(0/1)
Posaconazole	EUCAST	-	-	-	-	-	-	-	-	-	-	-	-
	CLSI	0.06	0.12	-	0.12	0.12	-	0.03	0.06	(0/2)	0.016	0.016	(0/1)
Itraconazole	EUCAST	-	-		-	-	-	-	-	-	0.047	0.047	-
	CLSI	0.06	0.12	(0/4)	0.12	0.12	33.3 (1/3)	0.12	0.12	(0/2)	0.03	0.03	(0/1)
Isavuconazole	EUCAST	-	-	-	-	-	-	-	-	-	-	-	-
	CLSI	0.06	0.12	(0/4)	0.12	0.12	33.3 (1/3)	-	-	-	0.008	0.008	(0/1)
Caspofungin	EUCAST	0.25	0.25	20 (1/5)	-	-	-	-	-	-	-	-	-
	CLSI	0.12	0.25	(0/5)	0.5	8	33.3 (2/6)	0.25	0.25	(0/2)	0.03	0.03	(0/1)
Micafungin	EUCAST	0.047	0.047	-	-	-	-	-	-	-	-	-	-
	CLSI	0.06	0.12	(0/5)	0.5	8	33.3 (2/6)	0.12	0.12	(0/2)	0.016	0.016	(0/1)
Anidulafungin	EUCAST	0.047	0.047	25 (1/4)	-	-	-	-	-	-	-	-	-
	CLSI	0.03	0.03	(0/4)	0.5	0.5	(0/3)	0.25	0.25	(0/2)	0.12	0.12	(0/1)
Amphotericin B	EUCAST	1	1	(0/9)	-	-	-	-	-	-	0.012	0.012	(0/1)
	CLSI	0.5	0.5	-	0.25	0.5	-	0.25	0.5	-	0.25	0.25	-

EUCAST resistance breakpoints according *Candida* species were as follows: *C. albicans*: fluconazole > 4 mg/L; voriconazole > 0.25 mg/L; posaconazole > 0.06 mg/L; itraconazole > 0.06 mg/L; micafungin > 0.016 mg/L; anidulafungin > 0.03 mg/L; caspofungin, if anidulafungin as well as micafungin resistant; amphotericin B > 1 mg/L; *C. parapsilosis* complex: fluconazole > 4 mg/L; voriconazole > 0.25 mg/L; posaconazole > 0.06 mg/L; itraconazole > 0.125 mg/L; micafungin > 2 mg/L; anidulafungin > 4 mg/L; caspofungin, if anidulafungin as well as micafungin resistant; amphotericin B > 1 mg/L; C. glabrata: fluconazole > 16 mg/L; micafungin > 0.016 mg/L; anidulafungin > 0.06 mg/L; caspofungin, if anidulafungin as well as micafungin resistant; amphotericin B > 1 mg/L; *C. tropicalis*: fluconazole > 4 mg/L; voriconazole > 0.25 mg/L; posaconazole > 0.06 mg/L; itraconazole > 0.125 mg/L; anidulafungin > 0.06 mg/L; caspofungin, if anidulafungin resistant; amphotericin B > 1 mg/L; *C. krusei*: anidulafungin > 0.06 mg/L; caspofungin, if anidulafungin resistant; amphotericin B > 1 mg/L; *C. dubliniensis*: fluconazole > 4 mg/L; voriconazole > 0.25 mg/L; posaconazole > 0.06 mg/L; amphotericin B > 1 mg/L; non-species related *Candida*: fluconazole > 4 mg/L. CLSI resistance breakpoints according *Candida* species were as follows: *C. albicans*: fluconazole > 4 mg/L; voriconazole > 0.5 mg/L; itraconazole > 0.5 mg/L; isavuconazole > 1 mg/L; micafungin > 0.5 mg/L; anidulafungin > 0.5 mg/L; caspofungin > 0.5 mg/L; *C. parapsilosis* complex: fluconazole > 4 mg/L; voriconazole > 0.5 mg/L; itraconazole > 0.5 mg/L; isavuconazole > 1 mg/L; micafungin > 4 mg/L; anidulafungin > 4 mg/L; caspofungin > 4 mg/L; *C. glabrata*: fluconazole > 32 mg/L; voriconazole > 0.5 mg/L, itraconazole > 0.5 mg/L, isavuconazole > 1 mg/L, micafungin > 0.125 mg/L; anidulafungin > 0.25 mg/L; caspofungin > 0.25 mg/L; *C. tropicalis*: fluconazole > 4 mg/L; voriconazole > 0.5 mg/L; itraconazole > 0.5 mg/L; isavuconazole > 1 mg/L; micafungin > 0.5 mg/L; anidulafungin > 0.5 mg/L; caspofungin > 0.5 mg/L; *C. krusei*: voriconazole > 0.5 mg/L; itraconazole > 0.5 mg/L; isavuconazole > 1 mg/L; micafungin > 0.5 mg/L; anidulafungin > 0.5 mg/L; caspofungin > 0.5 mg/L; *C. dubliniensis*: fluconazole > 4 mg/L; voriconazole > 0.5 mg/L; itraconazole > 0.5 mg/L; isavuconazole > 1 mg/L; micafungin > 0.5 mg/L; anidulafungin > 0.5 mg/L; caspofungin > 0.5 mg/L; *C. guilliermondii*: fluconazole > 4 mg/L; voriconazole > 0.5 mg/L; itraconazole > 0.5 mg/L; isavuconazole > 1 mg/L; micafungin > 4 mg/L; anidulafungin > 4 mg/L; caspofungin > 4 mg/L; CLSI resistance zone diameter breakpoints according to *Candida* species were as follows: *C. albicans*: fluconazole ≤13 mm; voriconazole ≤14 mm; caspofungin ≤14 mm; micafungin ≤19 mm; *C. parapsilosis* complex: fluconazole ≤13 mm; voriconazole ≤14 mm; caspofungin ≤10 mm; micafungin ≤13 mm; *C. glabrata*: fluconazole ≤14 mm; micafungin ≤27 mm; *C. tropicalis*: fluconazole ≤13 mm; voriconazole ≤14 mm; caspofungin ≤14 mm; micafungin ≤19 mm; *C. krusei*: voriconazole ≤12 mm; caspofungin ≤14 mm; micafungin ≤19 mm; *C. guilliermondii*: caspofungin ≤10 mm; micafungin ≤13 mm.

**Table 4 microorganisms-11-00560-t004:** Comparison per year of azole and echinocandin resistance in *Candida* species isolates tested in the study.

Year	Antifungal Resistance	Overall	*C. albicans*	*C.* Non-*albicans*	*Candida* spp. Excluding *C. krusei*	*C. parapsilosis* Complex	*C. glabrata*
2020	Fluconazole % (*n*)	11.5 (42/366)	0.7 (1/150)	19 (41/216)	10.7 (39/363)	22 (31/141)	15.2 (7/46)
	Voriconazole % (*n*)	4 (14/348)	3.5 (5/145)	4.4 (9/204)	4.1 (14/345)	5 (7/141)	5.9 (2/34)
	Echinocandin % (*n*)	2.6 (8/307)	0.8 (1/123)	3.8 (7/184)	2.3 (7/304)	0.8 (1/127)	7.5 (3/40)
2021	Fluconazole % (*n*)	14.2 (50/351)	**4.5** (8/176)	24 (42/175)	11.5 (39/340)	24.5 (23/94)	12.2 (6/49)
	Voriconazole % (*n*)	4.5 (15/331)	3.7 (6/161)	5.3 (9/170)	4.6 (15/324)	8.9 (8/90)	− (0/50)
	Echinocandin % (*n*)	2.5 (7/278)	− (0/136)	4.9 (7/142)	2.6 (7/273)	**5.1** (4/78)	− (0/53)

Numbers in bold indicate statistically significant difference between 2020 and 2021 (*p* ≤ 0.05).

## Data Availability

The dataset analyzed during the current study is available from the corresponding author upon reasonable request.

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
