# Peer review of "Multicentre Surveillance of Candida Species from Blood Cultures during the SARS-CoV-2 Pandemic in Southern Europe (CANCoVEU Project)"

_microorganisms, 2023, doi:10.3390/microorganisms11030560_

Round 1
Reviewer 1 Report
Overview
Boattini, et al. present an observational study examining Candida species and susceptibility data from six hospitals spread across five European countries from 2020-2021, coinciding with the COVID-19 pandemic. Given other reports of drug-resistant organism epidemiology changes during COVID, this is a worthwhile endeavor, and all the more so given that it was done as a collaboration across countries to give a more continental view of the situation and larger specimen numbers, opening the door to statistical examination. Please find more specific comments below:
Abstract
- L41-42: I would recommend rephrasing this sentence. As currently stated, it seems to imply nearly 100% of patients with critical conditions due to COVID-19 had antifungal resistance (i.e., “antifungal resistance was essentially present”).
Introduction
- It would be helpful for the reader to discuss a broad overview of COVID-19 epidemiology for the countries involved in this study particularly because much of the manuscript is comparing the full years of 2020 vs. 2021 but countries around the world saw the start of their COVID outbreak at different times in 2020. For instance, if some of these countries did not begin having large-scale COVID outbreaks in the area of their hospitals until spring 2020, it is not surprising that would find smaller numbers of Candida specimens in COVID-19 units in 2020 v. 2021 because the COVID units may not have existed for part of 2020. I’m flagging this for the introduction, but depending on how you choose to incorporate this, it may have a place in the methods section as well or instead of in the introduction.
Methods
- L115-152: This is quite a lot of space to take up and is challenging to read and compare in this text form. Could this information be presented in a table instead?
o This is especially lengthy give it is repeated as a footnote for table 3 later on.
o There are also no official CLSI amphotericin B breakpoints. Given this, the authors should explain how they decided on breakpoints for species-drug combinations without official breakpoints.
- Based on some of the author affiliations, I wondered whether pediatric patients or units were included in this analysis. Though patient demographics are not the focus of this study, there can be differences in adult vs. pediatric patient Candida epidemiology (e.g., C. parapsilosis outbreaks tend to happen in pediatric ICUs). If feasible, it would be great to see this clarified in the methods.
Results
- L162-164: It is not clear to me why the other 706 Candida blood cultures were excluded (from the 1452). Were these all because they were subsequent samples of the same species within 20 days? Or were some excluded due to not having AFST or being a species without set breakpoints? It may be helpful to add additional clarity here given the large proportion of samples that were excluded.
o I think this would especially be helpful because the discussion section includes how C. auris was not found but it is not clear until then that C. auris wasn’t simply excluded from analysis.
- Table 1: This table uses “trimester” as a column, but it is split into fourths, so should be quarters instead.
- Table 1: It seems like the rounding of decimals may be slightly off in this table. For example, 1/97 = 1.0% instead of the 1.1% listed. Or 39/107 = 36.4% instead of 36.5%. The authors should recheck these figures and update as needed.
- Table 1: For full transparency, I would recommend adding the p-values for all species, not just those found to be significant. Additionally, are the p-values the comparison of the total for 2020 v. 2021 or are you comparing at the quarterly figure trend? Having a designated labeled row for the p-value may help provide space to clarify these additional pieces.
- Table 2: Can we see similar clarification updates for this table surrounding the p-values be made as I’ve listed for table 1?
- L180-181: I initially had a difficult time understanding what was meant by “the increase of C. krusei was cumulative as no statistically significant differences were found between different departments.” I think the authors mean that while there was a statistical increase found in C. krusei overall for 2021 vs. 2020, they did not see anything or perhaps could not effectively compare C. krusei by both year and unit due to small numbers. I would recommend rephrasing or removing this part.
- L185-186: In the methods, the authors described how echinocandin resistance was classified (L153-154) but not how azole resistance was. In order to understand this result, the manuscript needs to include information on how resistance to the azole class as a whole was determined.
- L191-192: This sentence is missing the percent for voriconazole EUCAST.
Discussion
- Some of the conclusions are not reflected in the results presented. For example, lines 272-273 makes it sound as though there were statistically significant changes found in azoles overall, C. parapsilosis complex, and C. glabrata, but the only statistically significant changes found for 2020 v. 2021 for azoles was in C. albicans. Or the phrasing in lines 274-275 of echinocandin resistance “remain[ing] undetected] in 2021 implies that no such resistance was found in 2020, when in fact it was present in 2020. I would recommend the authors carefully review their discussion and conclusions phrasing broadly to ensure that the language is aligned with the results.
- L297-301: The authors present data showing enormous resistance to caspofungin in this study. I would not get the link in the citation to work, but found what looks to be the abstract here: https://academic.oup.com/mmy/article/60/Supplement_1/myac072P476/6706043. You can see in the full table presented in this link that caspofungin is highly different from the findings of anidulafungin and micafungin, both of which are known to be far more reliable indicators of echinocandin resistance than caspofungin (due to issues such as eagle effect). Because of this, I would recommend removing this reference or, at minimum, rephrasing to remove mention of the caspofungin results in this manuscript. Whenever possible, it would be best to stick to peer-reviewed full manuscripts rather than abstracts or poster citations.
- The discussion includes reference information about EUCAST v. CLSI, but I suspect this is only accounting for part of the differences. There was also a great range of susceptibility tested methods used in this study (Etest, VITEK 2, YeastOne, etc.) which likely also played a role in variability. The paper would be improved if more discussion about known differences between these methods and the associated limitations with analysis given the variation were included.
- Some of the discussion and conclusion sections include the hypothesis that some of the findings may have been influenced by changes in unit usage over time. Given that, I would not help wondering whether you had data on the total blood cultures during these time periods by unit, like you show for the Candida BC event numbers by unit. If that’s available, it may be worth looking into or presenting. If not, it may be worth quickly clarifying that that information was not accessible and is thus a study limitation.
Conclusions
- L349-350: This sentence is missing the percent resistant for C. albicans.
Please see some of the above conclusion notes in the discussion section.
General
- Check for italics and singular/plural consistency throughout the document.
Author Response
Turin, 17th February 2023
Editor-in-Chief
Microorganisms
We would like to thank the Editorial Team for his helpful suggestions, which in our view have enhanced the quality and strength of our study. We hope that in this revised version the manuscript is now suitable for publication in Microorganisms.
Please, note that the changes to the original manuscript have been highlighted in the text. The response to the Editor’s comments and ensuing modifications in the manuscript are also clearly indicated in the rebuttal.
Comments from Reviewers and point-by-point answers
Reviewer #1:
1) Boattini, et al. present an observational study examining Candida species and susceptibility data from six hospitals spread across five European countries from 2020-2021, coinciding with the COVID-19 pandemic. Given other reports of drug-resistant organism epidemiology changes during COVID, this is a worthwhile endeavor, and all the more so given that it was done as a collaboration across countries to give a more continental view of the situation and larger specimen numbers, opening the door to statistical examination.
We would like to thank the Reviewer for these accurate appraisals.
2) L41-42: I would recommend rephrasing this sentence. As currently stated, it seems to imply nearly 100% of patients with critical conditions due to COVID-19 had antifungal resistance (i.e., “antifungal resistance was essentially present”).
We would like to thank the Reviewer for this comment. Accordingly, the sentence was rephrased.
3) It would be helpful for the reader to discuss a broad overview of COVID-19 epidemiology for the countries involved in this study particularly because much of the manuscript is comparing the full years of 2020 vs. 2021 but countries around the world saw the start of their COVID outbreak at different times in 2020. For instance, if some of these countries did not begin having large-scale COVID outbreaks in the area of their hospitals until spring 2020, it is not surprising that would find smaller numbers of Candida specimens in COVID-19 units in 2020 v. 2021 because the COVID units may not have existed for part of 2020. I’m flagging this for the introduction, but depending on how you choose to incorporate this, it may have a place in the methods section as well or instead of in the introduction.
We would like to thank the Reviewer for this comment. To think of being able to date the start of the pandemic city by city (or centre by centre) is very difficult. In terms of diagnosis, the potentials have been very heterogeneous and hospitals have undergone absolutely non-uniform and non-standardised reorganisations, especially in the early 2020s. Accordingly, the following study limitation was added “In addition, the heterogeneity of the SARS-CoV-2 pandemic and the relative diagnostic capacities in the different centers even belonging to the same countries in the early 2020s and the continuous hospital reorganizations may have had a small influence on some of the results presented regarding the 2020 vs. 2021 epidemiological comparisons”.
3) L115-152: This is quite a lot of space to take up and is challenging to read and compare in this text form. Could this information be presented in a table instead? This is especially lengthy give it is repeated as a footnote for table 3 later on. There are also no official CLSI amphotericin B breakpoints. Given this, the authors should explain how they decided on breakpoints for species-drug combinations without official breakpoints.
We would like to thank the Reviewer for this comment. Accordingly, both EUCAST and CLSI breakpoints and CLSI amphotericin B breakpoints were removed from the text.
4) Based on some of the author affiliations, I wondered whether pediatric patients or units were included in this analysis. Though patient demographics are not the focus of this study, there can be differences in adult vs. pediatric patient Candida epidemiology (e.g., C. parapsilosis outbreaks tend to happen in pediatric ICUs). If feasible, it would be great to see this clarified in the methods.
We would like to thank the Reviewer for this comment. However, Candida spp positive blood culture events included in the study were coded and it is not possible to distinguish between adult and paediatric samples.
5) L162-164: It is not clear to me why the other 706 Candida blood cultures were excluded (from the 1452). Were these all because they were subsequent samples of the same species within 20 days? Or were some excluded due to not having AFST or being a species without set breakpoints? It may be helpful to add additional clarity here given the large proportion of samples that were excluded. I think this would especially be helpful because the discussion section includes how C. auris was not found but it is not clear until then that C. auris wasn’t simply excluded from analysis.
We would like to thank the Reviewer for this comment. These samples were excluded because they were considered to be related to the same positive blood culture event or subsequent samples, as reported in the methods.
6) Table 1: This table uses “trimester” as a column, but it is split into fourths, so should be quarters instead.
We would like to thank the Reviewer for this comment. Accordingly, trimester was corrected with quarter.
7) Table 1: It seems like the rounding of decimals may be slightly off in this table. For example, 1/97 = 1.0% instead of the 1.1% listed. Or 39/107 = 36.4% instead of 36.5%. The authors should recheck these figures and update as needed.
We would like to thank the Reviewer for this comment. The rounding of decimals was done to obtain 100% for each species. We would kindly point out to the reviewer that this rounding does not affect the results presented.
8) Table 1: For full transparency, I would recommend adding the p-values for all species, not just those found to be significant. Additionally, are the p-values the comparison of the total for 2020 v. 2021 or are you comparing at the quarterly figure trend? Having a designated labeled row for the p-value may help provide space to clarify these additional pieces.
We would like to thank the Reviewer for this comment. The reported p-values referred to the subtotal 2020 vs. 2021 comparison. According to the reviewer’s opinion, table 1 was revised and p-values of the comparison 2020 vs. 2021 were eliminated.
9) Table 2: Can we see similar clarification updates for this table surrounding the p-values be made as I’ve listed for table 1?
We would like to thank the Reviewer for this comment. Accordingly, table 2 was revised and comparison 2020 vs. 2021 eliminated.
10) L180-181: I initially had a difficult time understanding what was meant by “the increase of C. krusei was cumulative as no statistically significant differences were found between different departments.” I think the authors mean that while there was a statistical increase found in C. krusei overall for 2021 vs. 2020, they did not see anything or perhaps could not effectively compare C. krusei by both year and unit due to small numbers. I would recommend rephrasing or removing this part.
We would like to thank the Reviewer for this comment. Accordingly, the sentence was removed.
11) L185-186: In the methods, the authors described how echinocandin resistance was classified (L153-154) but not how azole resistance was. In order to understand this result, the manuscript needs to include information on how resistance to the azole class as a whole was determined.
We would like to thank the Reviewer for this comment. Accordingly, the definition of resistance to the azole class was added.
12) L191-192: This sentence is missing the percent for voriconazole EUCAST.
We would like to thank the Reviewer for this comment. Accordingly, it was added.
13) Some of the conclusions are not reflected in the results presented. For example, lines 272-273 makes it sound as though there were statistically significant changes found in azoles overall, C. parapsilosis complex, and C. glabrata, but the only statistically significant changes found for 2020 v. 2021 for azoles was in C. albicans. Or the phrasing in lines 274-275 of echinocandin resistance “remain[ing] undetected] in 2021 implies that no such resistance was found in 2020, when in fact it was present in 2020. I would recommend the authors carefully review their discussion and conclusions phrasing broadly to ensure that the language is aligned with the results.
We would like to thank the Reviewer for this comment. Accordingly, the sentences were rephrased as follows: “Resistance rate to azole over the study period increased significantly in C. albicans (fluconazole) and remained very high in C. parapsilosis complex” and “Resistance to echinocandin increased in 2021 especially in C. parapsilosis complex, while it was undetected in both C. albicans and C. glabrata during the same period.”
14) L297-301: The authors present data showing enormous resistance to caspofungin in this study. I would not get the link in the citation to work, but found what looks to be the abstract here: https://academic.oup.com/mmy/article/60/Supplement_1/myac072P476/6706043. You can see in the full table presented in this link that caspofungin is highly different from the findings of anidulafungin and micafungin, both of which are known to be far more reliable indicators of echinocandin resistance than caspofungin (due to issues such as eagle effect). Because of this, I would recommend removing this reference or, at minimum, rephrasing to remove mention of the caspofungin results in this manuscript. Whenever possible, it would be best to stick to peer-reviewed full manuscripts rather than abstracts or poster citations.
We would like to thank the Reviewer for this comment. The data presented in our study showed absolutely no high rates of caspofungin resistance. In discussion, we reported the most recently published IHMA data on two European and world surveillance programmes (all the links work). Although they are published in poster format, we are not convinced that the results will change after peer-review. Concerning our data, we reported that resistance to caspofungin was remarkable in C. tropicalis (10%) and C. krusei (20%) but the limited number of isolates limits the generalizability of these results and should be considered a study limitation.
15) The discussion includes reference information about EUCAST v. CLSI, but I suspect this is only accounting for part of the differences. There was also a great range of susceptibility tested methods used in this study (Etest, VITEK 2, YeastOne, etc.) which likely also played a role in variability. The paper would be improved if more discussion about known differences between these methods and the associated limitations with analysis given the variation were included.
We would like to thank the Reviewer for this comment. However, variability in the agreement between the commercial and reference methods for Candida species susceptibility was well made explicit in the discussion and has to be considered a study limitation.
16) Some of the discussion and conclusion sections include the hypothesis that some of the findings may have been influenced by changes in unit usage over time. Given that, I would not help wondering whether you had data on the total blood cultures during these time periods by unit, like you show for the Candida BC event numbers by unit. If that’s available, it may be worth looking into or presenting. If not, it may be worth quickly clarifying that that information was not accessible and is thus a study limitation.
We would like to thank the reviewer for this comment. The number of positive blood culture events for Candida species per unit was not available and should be considered a limitation of the study. Accordingly, we added the following limitation: "In addition, the heterogeneity of the SARS-CoV-2 pandemic and the relative diagnostic capacities in the different centers even belonging to the same countries in the early 2020s and the continuous hospital reorganizations may have had a small influence on some of the results presented regarding the 2020 vs. 2021 epidemiological comparisons".
17) L349-350: This sentence is missing the percent resistant for C. albicans. Please see some of the above conclusion notes in the discussion section.
We would like to thank the Reviewer for this comment. Accordingly, it was added.
18) Check for italics and singular/plural consistency throughout the document.
We would like to thank the Reviewer for this comment. Accordingly, the paper was largely revised.

Reviewer 2 Report
In this manuscript the authors describe the epidemiology of Candida spp bloodstream infections in six European tertiary hospitals during the first two years of SARS-CoV-2 pandemic, focusing on the species of Candida isolated and their antifungal susceptibility patterns. The aim of the study is well explained and attained with rigorous approach and the results are presented clearly and in detail. The topic is of great interest and concern, especially considering the fragmented data available so far and the lack of updated surveillance data on Candida BSI in the EU since the beginning of the pandemic. The findings are fairly well discussed and possible explanations are provided regarding the most remarkable differences identified in the epidemiology of Candida spp infections in different wards (medicine, surgery, non-COVID-19 ICU and COVID-19 ICU) and over the two-year period and some discrepancies in the susceptibility patterns following the EUCAST or CLSI breakpoints. Scarce relevance is given to the clinical data (host-related risk factors, hospital-related risk factors, comorbidities, complications and treatment) which are not of interest in terms of crude epidemiological assessment, nonetheless it could be worthwhile to investigate such parameters in order to better understand the epidemiological changes of Candida infections. A short paragraph depicting the study limitations has been included.
- I suggest underlying in the title and abstract that all the hospitals where the study was conducted belong to Southern Europe, thus possibly not representing thoroughly the overall European epidemiology.
- Even if no significant increase in overall Candida spp positive blood cultures events was observed during the study period, it may be helpful to estimate the incidence of Candida spp BSI per admissions in each setting over this time and compare it to the incidence in the pre-pandemic period. In fact, many departments (emergency departments, medical and surgical wards converted to COVID-19 wards, ICUs converted to COVID-19 ICUs) faced a substantial increase of the patient load while others had a decrease in the number of admissions: these trends have an impact on the denominator used to assess the incidence of Candida spp infections.
Author Response
Turin, 17th February 2023
Editor-in-Chief
Microorganisms
We would like to thank the Editorial Team for his helpful suggestions, which in our view have enhanced the quality and strength of our study. We hope that in this revised version the manuscript is now suitable for publication in Microorganisms.
Please, note that the changes to the original manuscript have been highlighted in the text. The response to the Editor’s comments and ensuing modifications in the manuscript are also clearly indicated in the rebuttal.
Comments from Reviewers and point-by-point answers
Reviewer #2:
1) In this manuscript the authors describe the epidemiology of Candida spp bloodstream infections in six European tertiary hospitals during the first two years of SARS-CoV-2 pandemic, focusing on the species of Candida isolated and their antifungal susceptibility patterns. The aim of the study is well explained and attained with rigorous approach and the results are presented clearly and in detail. The topic is of great interest and concern, especially considering the fragmented data available so far and the lack of updated surveillance data on Candida BSI in the EU since the beginning of the pandemic. The findings are fairly well discussed and possible explanations are provided regarding the most remarkable differences identified in the epidemiology of Candida spp infections in different wards (medicine, surgery, non-COVID-19 ICU and COVID-19 ICU) and over the two-year period and some discrepancies in the susceptibility patterns following the EUCAST or CLSI breakpoints. Scarce relevance is given to the clinical data (host-related risk factors, hospital-related risk factors, comorbidities, complications and treatment) which are not of interest in terms of crude epidemiological assessment, nonetheless it could be worthwhile to investigate such parameters in order to better understand the epidemiological changes of Candida infections. A short paragraph depicting the study limitations has been included.
We would like to thank the reviewer for these accurate appraisals.
2) I suggest underlying in the title and abstract that all the hospitals where the study was conducted belong to Southern Europe, thus possibly not representing thoroughly the overall European epidemiology.
We would like to thank the reviewer for this comment. Accordingly, title and abstract were revised.
3) Even if no significant increase in overall Candida spp positive blood cultures events was observed during the study period, it may be helpful to estimate the incidence of Candida spp BSI per admissions in each setting over this time and compare it to the incidence in the pre-pandemic period. In fact, many departments (emergency departments, medical and surgical wards converted to COVID-19 wards, ICUs converted to COVID-19 ICUs) faced a substantial increase of the patient load while others had a decrease in the number of admissions: these trends have an impact on the denominator used to assess the incidence of Candida spp infections.
We would like to thank the reviewer for this comment. This study was aimed at monitoring and comparing the epidemiology and antifungal susceptibility of Candida species isolated from blood cultures collected from patients admitted in six Southern European tertiary hospitals during the first two years of COVID-19 pandemic. Estimating the incidence of Candida spp positive blood culture event per admissions in each setting over this time and compare it to the incidence in the pre-pandemic period is out of the scope of the study.
